# Exogenous Serotonin Improves Salt Tolerance in Rapeseed (*Brassica napus* L.) Seedlings

**Yue Liu [1], Xiaoyu Ding [1], Yan Lv [1], Yong Cheng [1], Chunsheng Li [2], Lei Yan [1], Shanshan Tian [1] and Xiling Zou [1,\*]**

[1] Key Lab of Biology and Genetic Improvement of Oil Crops of Ministry of Agriculture and Rural Affairs, Oil Crops Research Institute of the Chinese Academy of Agricultural Sciences, Wuhan 430062, China; liuyue199212@163.com (Y.L.); dingxiaoyu1991@163.com (X.D.); lvyan322@163.com (Y.L.); chengyong58@139.com (Y.C.); yanlei2723@126.com (L.Y.); tianshanshan5020@163.com (S.T.)

[2] Hubei Institute of New Socialist Countryside Development, Hubei Engineering University, Xiaogan 432000, China; xgxh2006@163.com

\* Correspondence: zouxiling01@caas.cn; Tel.: +86-27-8682-4573

**Abstract:** Serotonin is a well-known agent that plays various roles in animals, and is little known in plants. In this study, the effect of exogenous serotonin was tested on *Brassica napus* L. (rapeseed) under salt stress. The results revealed that exogenous application of 200 μmol/L serotonin had the best protection under salinity. Exogenous serotonin effectively alleviated the growth inhibition of seedlings caused by salinity, and significantly promoted the accumulation of the fresh and dry weights of roots and shoots. Besides, although the $H_2O_2$ and malondialdehyde (MDA) contents were raised under salinity, they were reduced by exogenous serotonin. The chlorophyll content was decreased under salinity, and was increased by exogenous serotonin. Under salinity, serotonin effectively activated antioxidant enzyme system through improving the catalase (CAT), superoxide dismutase (SOD), and peroxidase (POD) activities, and the expression of *POD7*, *CAT3* and *Cu-SOD* genes was also up-regulated. The results also revealed exogenous serotonin increased the solute content by promoting the accumulation of soluble sugar and protein. In conclusion, salinity caused a toxicity to seedlings through oxidative damage to chlorophyll and cell membrane integrity, and serotonin possessed the ability of scavenging reactive oxygen species, osmotic pressure regulation and promoting growth, thus alleviating salinity of rape seedlings.

**Keywords:** serotonin; *Brassica napus* L. (rapeseed); salinity; seedling

## 1. Introduction

Salinity is an important abiotic stress to crop yield [1]. The stress leads to ionic, osmotic, secondary, and especially oxidative stress. Reactive oxygen species (ROS), including $H_2O_2$ and hydroxyl radical, are naturally generated by many cellular pathways [2], which interfere with normal metabolism by changing membrane lipids through peroxides. Under normal conditions, ROS are maintained at a low levels in chloroplast, mitochondria and peroxisome, and the rate of their production increases rapidly under adverse conditions [3]. Oxidative stress causes a significant increase in cell membrane peroxidation and malondialdehyde (MDA) content, which is harmful to plants. Under salinity, photosynthesis, osmoregulation, lipid metabolism and other major processes are affected [4,5], resulting plant growth inhibition and yield loss.

When plants are exposed to salt stress, ion compartmentalization occurs, which induces $Na^+/H^+$ transporter activity and scavenges ROS. In order to prevent the damage caused by reactive oxygen species, the plant antioxidant defense system plays a critical role in controling the ROS content, and peroxidase (POD), superoxide dismutase (SOD) and catalase (CAT) are important protective enzymes in enzymatic defense system. The activities of POD, SOD and CAT were increased [6,7], and the expressions of related genes were also enhanced under salinity [8–10]. Under salt stress, Osmolytes, such as sugars

and protein, are accumulated and external osmotic pressure are decreased. In order to maintain the balance of osmotic pressure in cells, osmotic adjustment substances such as soluble sugar (SS), soluble protein (SP) and proline (pro) are accumulated in plants prevent water leakage [1,10]. The accumulation of proline can significantly increase the tolerance of plants to salt stress [4]. The ability of salt tolerance involves regulation of ion and osmotic balance, as well as the repair effect on stress injury (detoxification), and some transgenic plants tolerant to salinity have been developed, with over-expressing genes related to salt absorption and transportation, ROS scavenging enzymes or compatible solute accumulation [11].

Exogenous hormones are extensively used to improve plant stress resistance, and it is vital to explore potential growth regulators and their mechanisms. Serotonin is an ubiquitous monoamine, and mediates a series of activities of various animal cells as a neurotransmitter, hormone and mitotic factor [12,13]. It is widely found in plants, and is also reported to be involved in a series of physiological functions related to germination, flowering, growth regulation, delay of senescence, and response to stresses [14–17]. Moreover, exogenous serotonin decreased the $H_2O_2$ and MDA content [18,19], and alleviated seedling growth inhibition under salt stress in sunflower (*Helianthus annuus* L.) [20].

Rapeseed (*Brassica napus* L.) is planted worldwide [21], and is greatly affected by salinity as other crop species. It is interesting to study whether serotonin could improve salt tolerance and explore the possible mechanism in rapeseed. Then, in this study, the effect of exogenous serotonin possible on rapeseed was studied under salinity by evaluating a number of phenotypic, physiological indices, antioxidant enzyme activities and gene expressions.

## 2. Materials and Methods

### 2.1. The Materials

ZS11, a rapeseed variety was used as the experimental material, and the experiment was carried out in the greenhouse.

### 2.2. Experimental Methods

Healthy seeds were sowed on the gauze that was submerged in 1/4 concentration of nutrient solution (NS). After 7 days, the seedlings were evenly selected and planted in the pot with modified Hogland nutrient solution [22], and each pot contained 24 seedlings. Then, the seedlings were hydroponically cultured in the growth chamber, and the nutrient solution was renewed every other day.

One week later, uniform seedlings with two leaves were selected for the following treatments: (1) only NS (The control, C); (2) NS + 0.75% NaCl (NaCl treatment, N); (3) NS + 0.75% NaCl + 50 μmol/L serotonin (50 Ser); (4) NS + 0.75% NaCl + 100 μmol/L serotonin (100 Ser) (5) NS + 0.75% NaCl + 200 μmol/L serotonin (200 Ser); (6) NS + 0.75% NaCl + 300 μmol/L serotonin (300 Ser).

### 2.3. The Measurements

After 7 days of treatment, seedlings were randomly selected from each group, and the length and fresh weight of the root and shoot of each plant were measured. There were 3 biological repeats in all treatments, and each replicate included 8 seedlings.

The content of chlorophyll was determined by a SPAD meter (SPAD-502, Minota, tokyo, Japan), and the same position of three seedlings was measured in each replication.

After one day of treatment, the third leaf was collected and stored at −80 °C, and the biochemical indexes of POD, SOD, $H_2O_2$, MDA, PRO, soluble protein (SP), soluble sugar (SS) were measured as previously mentioned [23]. Serotonin was purchased from Shanghai Yuanye Biotechnology Co., Ltd. All the biochemical indicators of leaves were determined using commercial kits (Jiancheng, Nanjing, China).

*2.4. The RT-qPCR*

The third leaf of the seedlings was taken at 1, 2, 4, 8, 12 and 24 h after treatment, and the control samples were collected at the same time. HiPure Total RNA Mini Kit (Magen Biotech Co., Ltd., Beijing, China) was used for RNA extraction, and a cDNA synthesis Kit (Thermo Fisher Scientific, Rockford, IL, USA) was used for reverse transcription. The expression of actin was used as control. Primers were designed by Primer Premier 5 (Table 1). The reagents, procedures and instruments of RT qPCR were followed as described by Yan et al. [24]. The calculation was performed using $2^{-\Delta\Delta CT}$ method, and the standard error between three biological copies of each experiment was calculated.

**Table 1.** The list of primers for qRT-PCR of Rapeseed Seedlings.

| Gene | Accession Number | Forward Primer | Reverse Primer |
|---|---|---|---|
| *POD7* | XM_013819255.1 | CCGTGGCTACAGCAACAGTC | CTCACCGCTACTACCAGTCAAGG |
| *CAT3* | XM_013786778.1 | GAAGGCTTCGGTGTCCACAC | GGCTCCACCAACCACCTTT |
| *Cu-SOD* | XM_013808924.1 | GGTAACGGATCTGATTGGACG | ACGGTGGTTCCATCACAAGAA |
| *Actin* | AF111812 | CTCCTGAAGAGCACCCTG | CCCTCGTAGATTGGCACA |

*2.5. Statistical Analysis*

Data were subjected to analysis of variance (ANOVA); mean separation was performed using the least significance difference (Duncan, $p < 0.05$) procedure from the SPSS statistical package ((International Business Machines Corporation, Armonk, NY, USA). Graphs were plotted using Origin 8.0 (OriginLab Corporation, Northampton, MA, USA).

## 3. Results

*3.1. Effects of Exogenous Serotonin on Phenotypical of B. napus under Salinity*

After 7 days of NaCl treatment, the seedlings showed typical symptoms of salt poisoning, such as yellow, rolled and smaller leaves in plants (Figure 1). These symptoms were alleviated by adding 50, 100, 200 or 300 μmol/L serotonin. When the concentration of serotonin was 50–200 μmol/L, the growth of rape seedlings gradually improved under salt stress. However, when 300 μmol/L serotonin was added, the growth vigor of rape seedlings was significantly weaker than that of 200 μmol/L serotonin.

*3.2. Effects of Exogenous Serotonin on the Growth under Salinity*

As shown in Table 2, all the characters were significantly reduced by salinity. Exogenous serotonin promoted the growth of seedlings under salinity, and 200 μmol/L of exogenous serotonin increased the fresh and dry weight of the shoot and root by 43.4%, 43.4%, 48.5%, and 19.5% compared to that without serotonin under salinity, respectively. However, the root length decreased slightly after serotonin was applied. There was no significant difference in the shoot length of seedlings among the treatments.

**Table 2.** Effect of different concentrations of serotonin on various physiological indicators in rapeseed seedlings under salt stress.

| Treatment | Root Length(cm) | Shoot Length (cm) | Shoot Fresh Weights (g) | Shoot Dry Weight (g) | Root Fresh Weights (g) | Root Dry Weight (g) |
|---|---|---|---|---|---|---|
| C | 34.50 ± 0.34 [a] | 3.01 ± 0.04 [a] | 3.5608 ± 0.1746 [a] | 0.2290 ± 0.005 [a] | 0.4006 ± 0.0103 [b] | 0.0359 ± 0.0027 [ab] |
| N | 33.18 ± 0.33 [b] | 2.98 ± 0.05 [a] | 1.5218 ± 0.301 [d] | 0.1172 ± 0.003 [d] | 0.2915 ± 0.003 [d] | 0.0333 ± 0.0015 [b] |
| 50 Ser | 30.87 ± 0.38 [c] | 2.24 ± 0.03 [d] | 1.8342 ± 0.1344 [c] | 0.1474 ± 0.0024 [c] | 0.3786 ± 0.0077 [b] | 0.0324 ± 0.0007 [bc] |
| 100 Ser | 28.87 ± 0.5 [d] | 2.83 ± 0.05 [ab] | 2.1052 ± 0.0012 [bc] | 0.1639 ± 0.0046 [b] | 0.4002 ± 0.004 [b] | 0.0323 ± 0.0009 [bc] |
| 200 Ser | 26.75 ± 0.45 [e] | 2.63 ± 0.03 [bc] | 2.1825 ± 0.0009 [b] | 0.1681 ± 0.0025 [b] | 0.4330 ± 0.011 [a] | 0.0398 ± 0.0004 [a] |
| 300 Ser | 24.92 ± 0.26 [f] | 2.55 ± 0.16 [c] | 1.8171 ± 0.029 [c] | 0.1448 ± 0.0023 [c] | 0.3290 ± 0.0113 [c] | 0.0285 ± 0.0005 [c] |

Note: C, N, 50 Ser, 100 Ser, 200 Ser and 300 Ser represent the seedling were treated by 0 NaCl and 0 serotonin (C), 0.75% NaCl and 0 serotonin (N), 0.75% NaCl and 50 μmol/L serotonin (50 Ser), 0.75% NaCl and 100 μmol/L serotonin (100 Ser), 0.75% NaCl and 200 μmol/L serotonin (200 Ser), 0.75% NaCl and 300 μmol/L serotonin (300 Ser) respectively. Date represent mean ± SD, *n* = 3. Different letters, a, b, c, d, e, and f, indicate significant differences ($p < 0.05$).

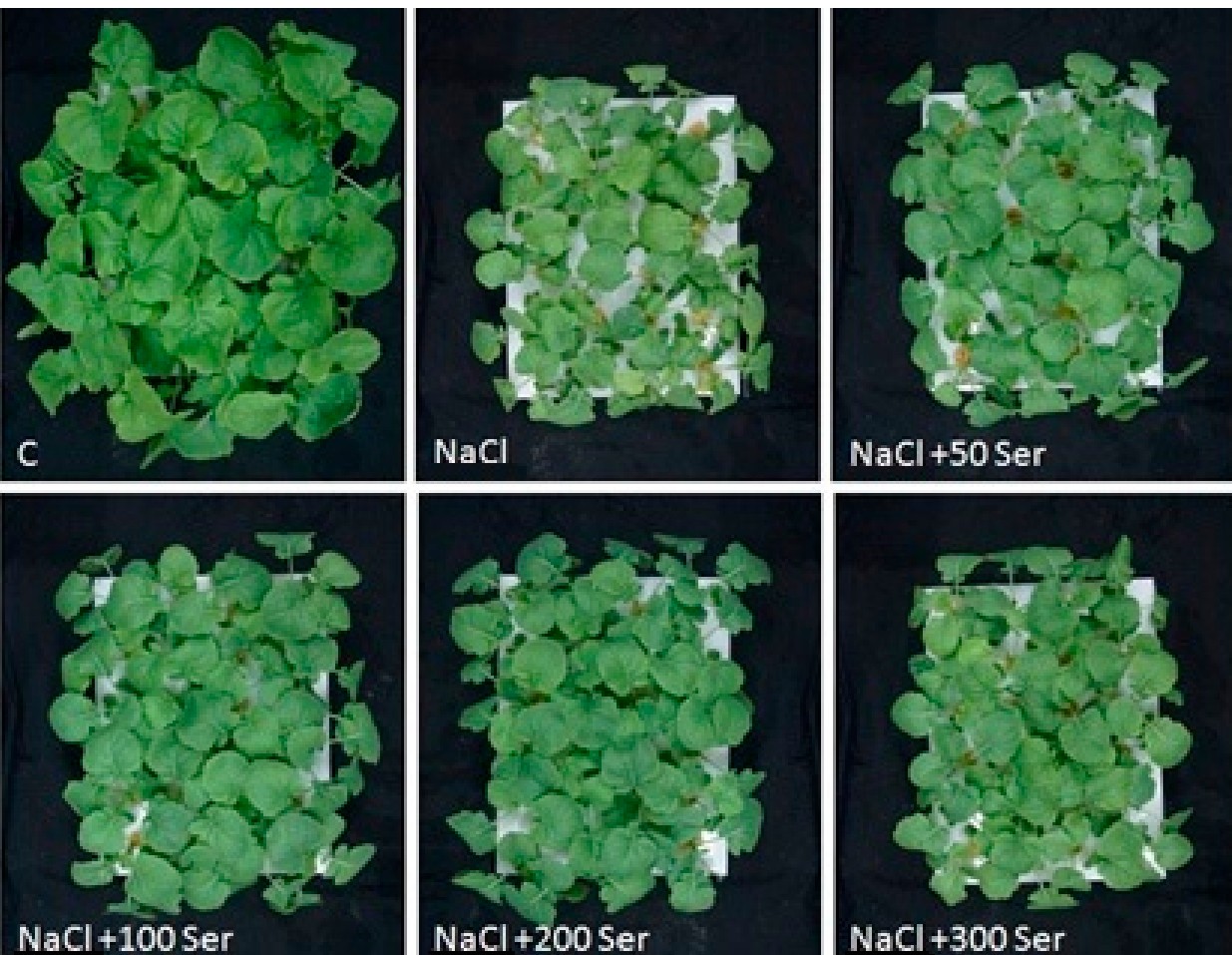

**Figure 1.** Exogenous serotonin alleviated salinity of rapeseed seedlings. C, NaCl, NaCl + 50 Ser, NaCl + 100 Ser, NaCl + 200 Ser and NaCl + 300 Ser represent the seedling were treated by 0 NaCl and 0 serotonin (C), 0.75% NaCl and 0 serotonin (NaCl), 0.75% NaCl and 50 μmol/L serotonin (NaCl + 50 Ser), 0.75% NaCl and 100 μmol/L serotonin (NaCl + 100 Ser), 0.75% NaCl and 200 μmol/L serotonin (NaCl + 200 Ser), 0.75% NaCl and 300 μmol/L serotonin (NaCl + 300 Ser) respectively. The three-week-old seedlings are shown above: two-week-old seedlings after one week of treatment.

### 3.3. Effects of Exogenous Serotonin on MDA, $H_2O_2$ and Chlorophyll Contents under Salinity

The content of MDA was significantly increased by 319.7% under salinity, and it was decreased by exogenous serotonin (Figure 2A). The change trend of the $H_2O_2$ content was similar to that of the MDA content. Under salinity, the $H_2O_2$ content in leaves was increased by 26.9% compared with the control, and that of seedlings treated with 50, 100, 200 and 300 μmol/L serotonin was decreased by 11.8%, 12.7%, 14.6% and 4.5% compared with that under salt stress, respectively (Figure 2B). In addition, the chlorophyll content of leaves was decreased by 47.9% under salinity, but was increased by serotonin (Figure 2C). The chlorophyll accumulation of the seedlings with 300 μmol/L of serotonin was lower than that of the seedlings with 200 μmol/L serotonin.

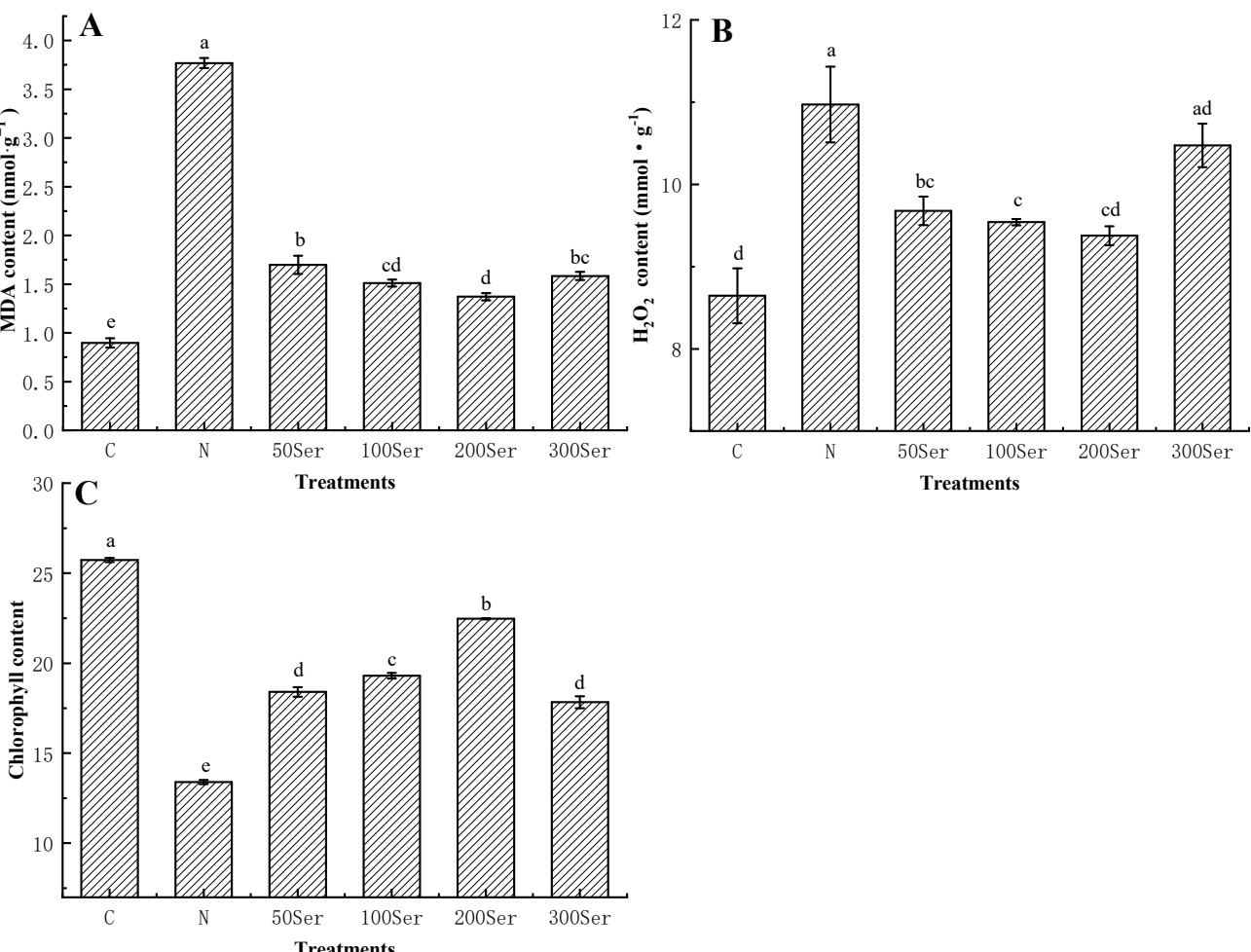

**Figure 2.** Effects of different concentrations of serotonin on the MDA, H$_2$O$_2$, Chlorophyll contents under salt stress. (**A**), the malonyldialdehyde (MDA) content; (**B**), the hydrogen peroxide (H$_2$O$_2$) content; (**C**), the Chlorophyll content. C, N, 50 Ser, 100 Ser, 200 Ser and 300 Ser represent the seedling were treated by 0 NaCl and 0 serotonin (C), 0.75% NaCl and 0 serotonin (N), 0.75% NaCl and 50 μmol/L serotonin (50 Ser), 0.75% NaCl and 100 μmol/L serotonin (100 Ser), 0.75% NaCl and 200 μmol/L serotonin (200 Ser), 0.75% NaCl and 300 μmol/L serotonin (300 Ser) respectively. Error bars represent SD (*n* = 3). Different letters, a, b, c, d, and e, indicate significant differences (*p* < 0.05).

### 3.4. Effects of Exogenous Serotonin on Osmoregulatory Substances under Salinity

Compared to that of the control, the SS content in leaves under salinity was significantly increased (Figure 3A), and exogenous serotonin had a significant positive effect under salinity. The highest SP content was found in the seedlings exposed to salinity with 200 μmol/L of exogenous serotonin. However, compared with 200 μmol/L serotonin, the use of 300 μmol/L serotonin reduced SP content by 8.0% (Figure 3B). Under salinity, the proline content was increased by 1308.0% under salinity (Figure 3C). There were no differences between the proline content of the NaCl and that with 50 or 100 μmol/L serotonin. Compared with that under salt stress, the proline content was increased by 13.9% with 200 μmol/L serotonin, and was decreased by 28.1% when exogenous serotonin was added at 300 μmol/L.

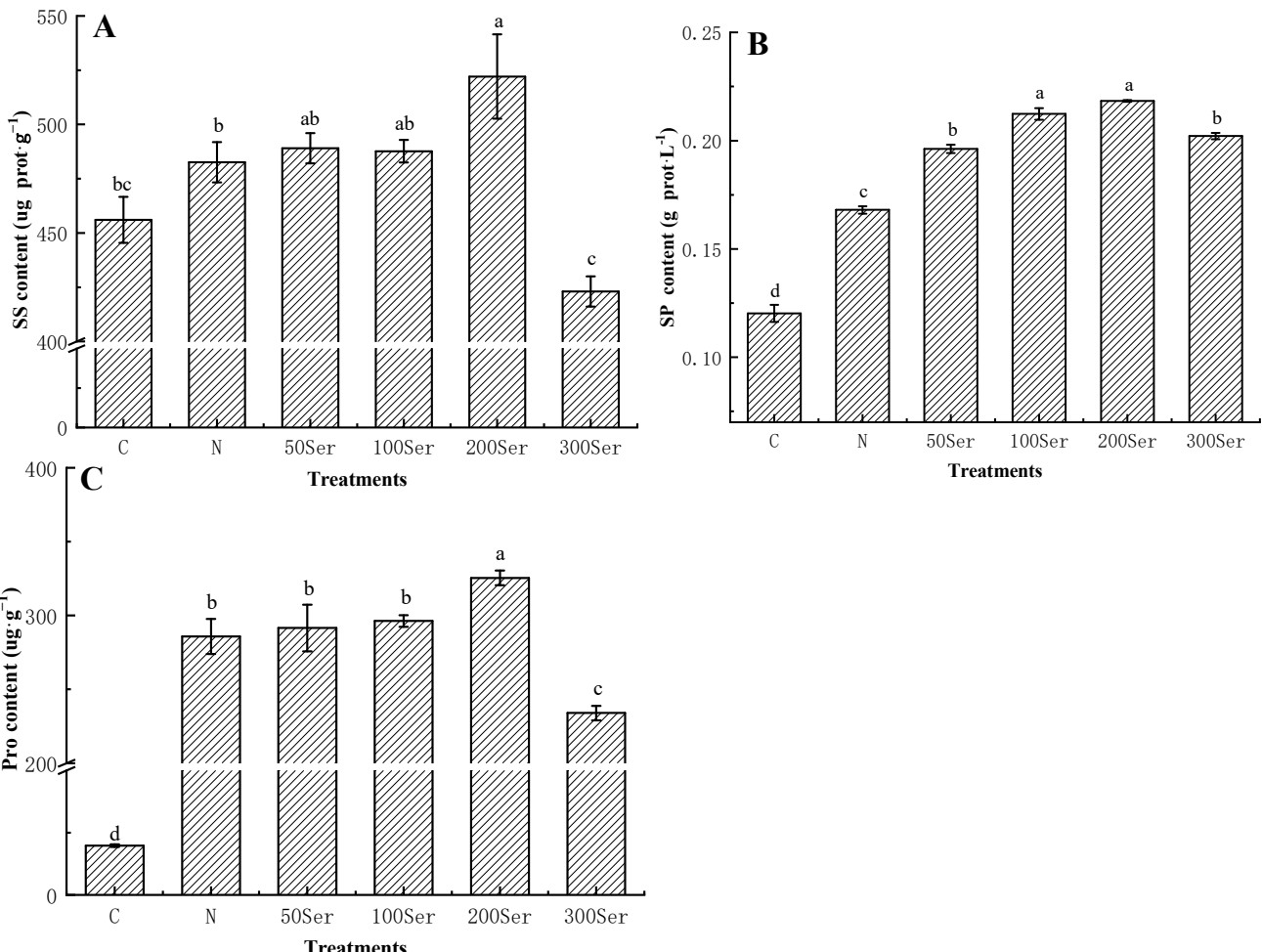

**Figure 3.** Effects of different concentrations of serotonin on the osmolyte content under salt stress. (**A**), total soluble sugar (SS) content. (**B**), total soluble protein (SP) content. (**C**), proline (Pro) content. C, N, 50 Ser, 100 Ser, 200 Ser and 300 Ser represent the seedling were treated by 0 NaCl and 0 serotonin (C), 0.75% NaCl and 0 serotonin (N), 0.75% NaCl and 50 μmol/L serotonin (50 Ser), 0.75% NaCl and 100 μmol/L serotonin (100 Ser), 0.75% NaCl and 200 μmol/L serotonin (200 Ser), 0.75% NaCl and 300 μmol/L serotonin (300 Ser) respectively. Error bars represent SD (*n* = 3). Different letters, a, b, c, and d, indicate significant differences (*p* < 0.05).

### 3.5. Effects of Exogenous Serotonin on Antioxidant Enzymes under Salinity

The POD activity was significantly increased under salinity (Figure 4A), and was further enhanced by exogenous application of serotonin. The POD activity was the highest with 200 μmol/L of serotonin, and was decreased by 300 μmol/L of serotonin. The CAT activity of seedlings was increased under salinity, and 50, 100, 200 and 300 μmol/L of exogenous serotonin further increased the CAT activity by 10.0%, 11.3%, 24.9% and 3.4% compared with that under salt stress, respectively (Figure 4B). A similar trend was also found in the SOD activity under the treatments (Figure 4C). The SOD activity was increased by 3.4% under the stress, while it was further increased by 1.9%, 5.2%, 10.6% and 3.1% compared with that under salt stress with the application of 50, 100, 200, and 300 μmol/L of exogenous serotonin, respectively.

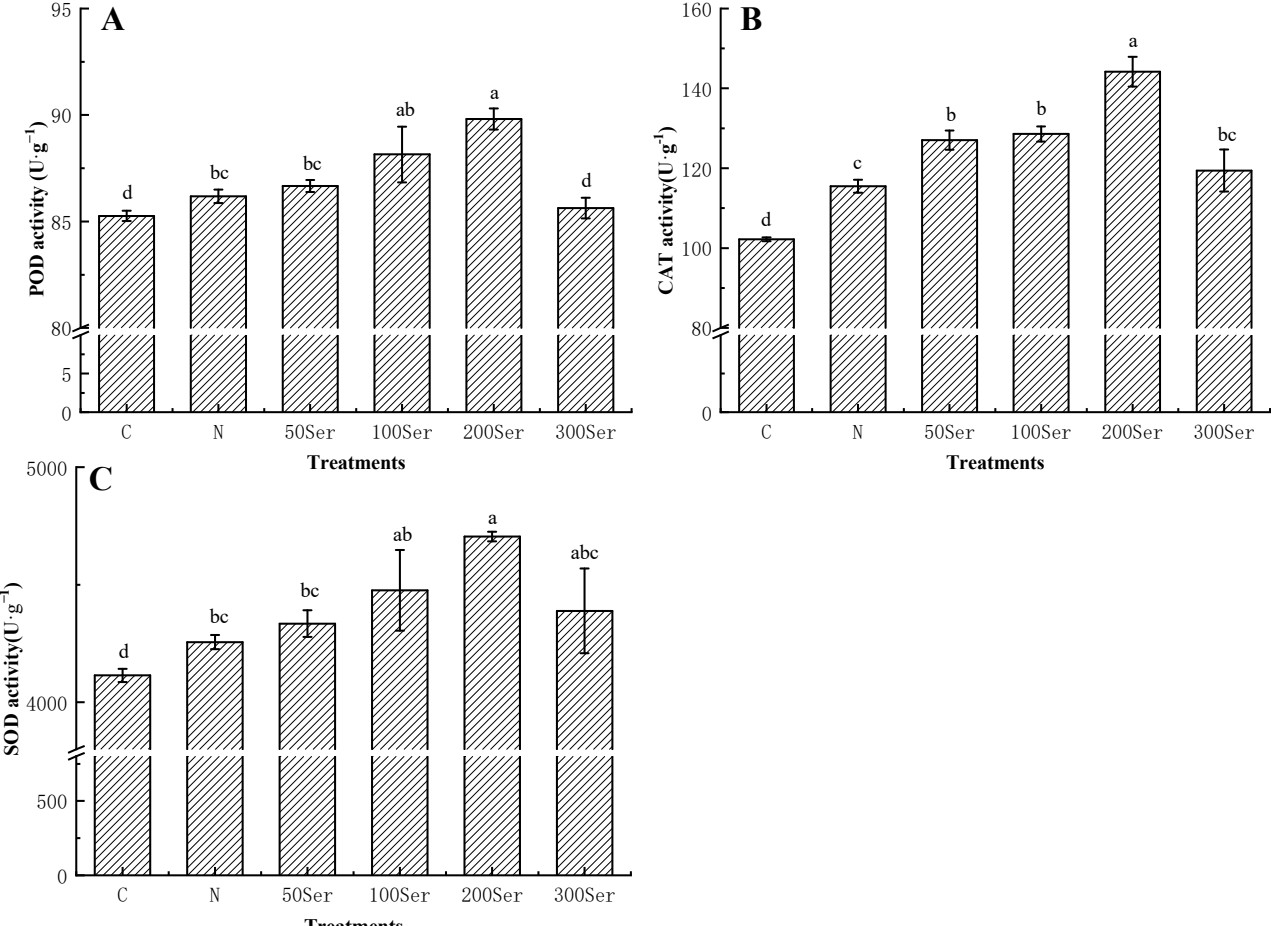

**Figure 4.** Effects of different concentrations of serotonin on the antioxidant enzymatic activities under salt stress. (**A**), the peroxidase (POD) activities. (**B**), the catalase (CAT) activities. (**C**), the superoxide dismutase (SOD) activities. C, N, 50 Ser, 100 Ser, 200 Ser and 300 Ser represent the seedling were treated by 0 NaCl and 0 serotonin (C), 0.75% NaCl and 0 serotonin (N), 0.75% NaCl and 50 μmol/L serotonin (50 Ser), 0.75% NaCl and 100 μmol/L serotonin (100 Ser), 0.75% NaCl and 200 μmol/L serotonin (200 Ser), 0.75% NaCl and 300 μmol/L serotonin (300 Ser) respectively. Error bars represent SD ($n = 3$). Different letters, a, b, c, and d, indicate significant differences ($p < 0.05$).

The expression patterns of the *POD7*, *CAT3*, and *Cu-SOD* genes were detected at 1 h, 2 h, 4 h, 8 h, 12 h and 24 h of the treatment. The expressions of *POD7*, *CAT3*, and *Cu-SOD* were down-regulated at 1h of salinity, and were up-regulated a few hours later. The expression of *POD7* was up-regulated at 2 h, 4 h, 8 h, 12 h and 24 h of salinity, and 200 μmol/L of exogenous serotonin up-regulated the expression level of *POD7* at each time point compared to that of the control or salinity (Figure 5A). Expression of *CAT3* was up-regulated at 12 h and 24 h under salinity, and the expression of *CAT3* was further up-regulated by exogenous serotonin at 12 h and 24 h (Figure 5B). The expression of *Cu-SOD* was up-regulated by 8 h and 12 h of salinity and application of exogenous serotonin (Figure 5C).

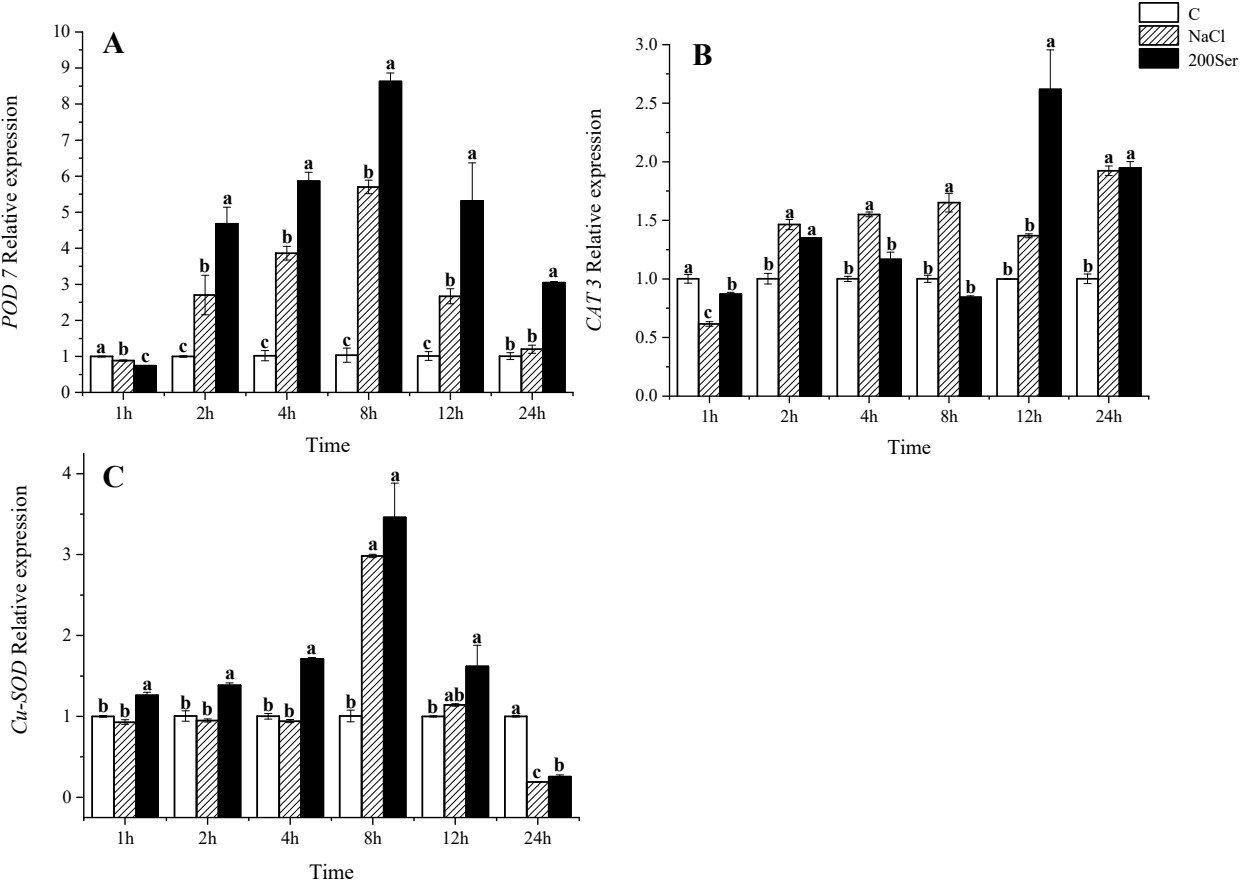

**Figure 5.** Effects of 200 μmol/L serotonin on the relative expression analysis of antioxidant enzyme genes under salt stress. (**A**–**C**), the transcriptional levels of genes *POD7* (XM_013819255.1), *CAT3* (XM_013786778.1), *Cu-SOD* (XM_013808924.1) were analyzed by real-time quantitative PCR (RT-qPCR). 0 NaCl and 0 serotonin (C), 0.75% NaCl and 0 serotonin (NaCl), 0.75% NaCl and 200 μmol/L serotonin (200 Ser). Error bars represent SD (*n* = 3). Different letters, a, b and c, indicate significant differences (*p* < 0.05).

## 4. Discussion

Salinity severely inhibits plant growth, and it is important to improve stress tolerance of crops [25,26]. Therefore, the effects of exogenous serotonin on rapeseed seedlings under salinity were studied in this study. The phenotypical data showed that salinity significantly inhibited plant growth (Figure 1 and Table 2), and resulted in an obvious decrease in the fresh and dry weight of plants. Application of exogenous serotonin noticeably alleviated the negative effect of salinity and promoted the seedling growth. Our results showed exogenous serotonin significantly increased the root weight, but not the root length. Therefore, serotonin may be involved in the formation of lateral roots as that in *Arabidopsis* [27]. With the increase of concentration, the alleviating effect of serotonin on salt stress gradually increased, but when the concentration reached 300 μmol/L, the alleviating effect began to weaken obviously, which indicated that high concentration of serotonin not only could not alleviate salt stress, but also could produce ion toxicity to plants [10].

Salinity affected the photosynthetic capacity of photosynthetic organs and decreased photosynthetic characters in plants [28]. Under stress, the chlorophyll decrease has always been an adverse effect on crop growth [28,29]. In our study, exogenous serotonin significantly increased the chlorophyll concentration, which may be one of the reasons for promoting seedling growth. Salt stress may affect photosynthesis through osmotic stress, ion toxicity and accumulation of photosynthetic products. Meanwhile, it also affects the process of light absorption, electron transfer and carbon assimilation. Studies have shown

that salt stress caused severe damage to chloroplast layer structure and rupture of plasma membrane, which affected plant growth and development under photosynthesis [2,28]. Abiotic stress leads to excessive ROS production in plant cells, and salinity can produce a large number of ROS [30]. $H_2O_2$ is one of ROS, and the high concentration of $H_2O_2$ can cause lipid peroxidation and membrane permeability changes [31,32]. In this study, $H_2O_2$ was increased under salinity, and the addition of exogenous serotonin inhibited the increase of $H_2O_2$ (Figure 2B). MDA is an end-product of lipid peroxidation, indicating the degree of membrane lipid peroxidation [33]. MDA affects the function and activity of nucleic acid, protein and enzyme, and even makes them lose their function [4,5]. Our results revealed that salinity significantly increased the MDA content, while exogenous serotonin significantly decreased the MDA content compared with that under salt stress (Figure 2A). Salinity damaged cell membrane and increased the MDA content, and serotonin improves salt tolerance by reducing the MDA content under salinity. It was also indicated that the membrane lipid peroxidation of plants was enhanced under salt stress, resulting in the increase of the MDA content. The addition of serotonin alleviated the membrane lipid peroxidation under salt stress [34].

In order to reduce oxidative damage and maintain cell metabolic function, it is necessary to keep a balance between ROS production and degradation under stress, and the antioxidant system can control ROS levels in plants [35]. When seedlings are exposed to salinity, ROS accumulated in plants could be removed through the synergistic action of protective enzymes (SOD, POD and CAT) [36]. In this study, these enzyme activities were enhanced under salinity (Figure 4). Exogenous application of serotonin further increased the activity of POD, SOD and CAT, resulting the decrease of the ROS content. Previous reports have reported serotonin had an anti-age effect by effectively eliminating ROS [16]. Meanwhile, the qPCR results revealed that exogenous serotonin significantly up-regulated the expression of *POD7*, *CAT3*, and *Cu-SOD* (Figure 5), indicating serotonin stimulated the activities of the main antioxidant enzymes by up-regulating the expression of corresponding genes, thus protecting cells under adverse conditions. This study indicated that there are two ways for serotonin in protecting the plants from injuries induced by oxidative stress under salinity. On the one hand, serotonin might act as a free radical scavenger indepdent of the antioxidant system [18,19]. On the other hand, it can enhance the gene expression and the activity of antioxidant enzymes [20].

Another major factor of salt tolerance is the ability of osmolyte accumulation to regulate cell osmotic pressure. The main osmolytes include free proline, soluble sugar and protein, and their content is also indicators of stress tolerance. Proline is reported to be accumulated as a compatible osmolyte mediating osmotic adjustment under salinity [37–39]. Besides, proline also play a role in detoxifying by scavenging ROS in injured cells [40]. In our study, the content of proline was increased under salinity, and was further enhanced by serotonin (Figure 3C). In addition, other studies also have shown that SS and SP played important roles in both osmoregulatory and active oxygen scavenging system [21,41,42]. In this study, exogenous application of serotonin also increased the content of SS and SP under salinity (Figure 3A,B).

## 5. Conclusions

Oxidative damage by salinity resulted in chlorophyll damage and decreased cell membrane integrity, which inhibited the growth of rapeseed seedlings. Exogenous serotonin has the function of antioxidation, regulating osmotic activity and promoting growth, which could effectively improve salt tolerance of plants.

**Author Contributions:** X.Z. conceived of the study and participated in its design. Y.L. (Yue Liu) carried out most of the work and drafted the manuscript. C.L. was involved in designing and planning the work, and in interpreting the results. Y.C. and X.D. prepared the seeds and helped with the data analysis. L.Y. and S.T. participated the assessment of physiological indicators. Y.L. (Yan Lv) helped to revise the English in this article. All authors have read and agreed to the published version of the manuscript.

**Funding:** This work was supported by the National Key R&D Program of China (2018YFD0100905); Hubei Special Project of Central Dovernment Guiding Local Science and Technology Development (2019ZYYD038); The Agricultural Science and Technology Innovation Program (ASTIP); The Hubei Agricultural Science and Technology Innovation Center (2016-620-000-001-048).

**Institutional Review Board Statement:** Not applicable.

**Informed Consent Statement:** Not applicable.

**Data Availability Statement:** The data presented in this study are available on request from the corresponding author.

**Conflicts of Interest:** The authors declare no conflict of interest.

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
