# Peer review of "Exogenous Serotonin Improves Salt Tolerance in Rapeseed (Brassica napus L.) Seedlings"

_agronomy, doi:10.3390/agronomy11020400_

Round 1

Reviewer 1 Report

In this manuscript, 'Exogenous serotonin improves salt tolerance in rapeseed (Brassica napus L.) seedlings,' the authors described the increase of salinity stress tolerance by using exogenous serotonin treatment the rapeseed seedlings. Although the authors did their best to describe the methods used and the results obtained, some analyses are required for the manuscript. See the comments below. 

  1. Provide more detailed information of the Seratonin background in the introduction part, because Seratonin is widely studied in a human as a mood controlling hormone, but a few in the plant field. And add the reason why you are interested in Seratonin in plants for abiotic stress response. 
  2. line 77; Did you analyze only 8 seedings in each measurement? I believe, 7-day old seeding was not big enough to measure the dry weight of every single plant. Provide a more detailed method in section 2.3. 
  3. Add more detailed the figure legend for each figure
  4. Combine the same group of figures in the results part. Figure 2, 3, and 4 in one figure. figure 5, 6, 7 in one figure, figure 8, 9, 10 in one figure. Because the many figures are not easy to read and understand. 

Author Response

In this manuscript, 'Exogenous serotonin improves salt tolerance in rapeseed (Brassica napus L.) seedlings,' the authors described the increase of salinity stress tolerance by using exogenous serotonin treatment the rapeseed seedlings. Although the authors did their best to describe the methods used and the results obtained, some analyses are required for the manuscript. See the comments below. 

  1. Provide more detailed information of the Serotonin background in the introduction part, because Serotonin is widely studied in a human as a mood controlling hormone, but a few in the plant field. And add the reason why you are interested in Seratonin in plants for abiotic stress response.

Reply:It has been revised based on the suggestion.

  1. line 77; Did you analyze only 8 seedlings in each measurement? I believe, 7-day old seeding was not big enough to measure the dry weight of every single plant. Provide a more detailed method in section 2.3. 

Reply:The seeds were germinated in 1/4 Hogland nutrient solution for one week, then the similar seedlings were retained. The seedlings were cultured for another week, and then were treated with stress. The seedlings were measured after one week of stress treatment. Therefore, we actually measured three week old seedlings. Each group had 3 replicates and each replicate had 8 seedlings (24 seedlings were measured in each group), which was explained in Section 2.2.

  1. Add more detailed the figure legend for each figure.

Reply: It has been revised based on the suggestion.

  1. Combine the same group of figures in the results part. Figure 2, 3, and 4 in one figure. figure 5, 6, 7 in one figure, figure 8, 9, 10 in one figure. Because the many figures are not easy to read and understand. 

Reply: It has been revised based on the suggestion.

Reviewer 2 Report

The subject of the manuscript entitled “Exogenous serotonin improves salt tolerance in rapeseed (Brassica napus L.) seedlings” fits the profile of Agronomy journal. The study delivers some interesting results and can be a source of valuable information. However, the authors made shortcomings that should be corrected and/or revised. 

  1. Line 39-47, authors have to comprehensively review the tolerance mechanism against salt stress in plants.
  2. Line 48-58, authors should cite more previous studies of the protective effect of serotonin in stress and/or the stress physiology of rapeseed.
  3. The hypothesis and the object of this study should be described.
  4. Serotonin is the most important chemical in this study, but authors did not mention the chemical source of serotonin in M&M.
  5. Table 1 should be next to 2.4 of M&M, and the accession number of each gene should be provided.
  6. Line 100, how did authors know the seedlings treated with 200 per was the best? According to Figure 1, I think seedlings treated with 100 and 200 per were similar.
  7. Decimal places should be revised moderately in Table 2.
  8. Line 117, should be H2O2.
  9. SPAD value is just a value without any unit in Figure 4.
  10. Line 147, proline content of the control group was significantly lower than that with 50 or 100 serotonin.
  11. Authors have to conduct statistical analysis for data of gene expression and mark significant differences in Figure 11-13.
  12. Figure 11, should be 200Ser
  13. Line 201, Arabidopsis should be in italic.
  14. The discussion of this manuscript is short and weak. Authors should cite more previous studies to support their results, hypothesis, or theory.
  15. Line 226, I didn't figure out which two ways.
  16. Figure 2-4, 5-7, 8-10, and 11-13 would be combined as a figure with multiple panels, respectively.

Author Response

The subject of the manuscript entitled “Exogenous serotonin improves salt tolerance in rapeseed (Brassica napus L.) seedlings” fits the profile of Agronomy journal. The study delivers some interesting results and can be a source of valuable information. However, the authors made shortcomings that should be corrected and/or revised. 

  1. Line 39-47, authors have to comprehensively review the tolerance mechanism against salt stress in plants.

Reply: It has been revised.

  1. Line 48-58, authors should cite more previous studies of the protective effect of serotonin in stress and/or the stress physiology of rapeseed.

Reply: Previous studies about the effects of serotonin on plant stress are very few, and almost all of them have been cited.

  1. The hypothesis and the object of this study should be described.

Reply: It has been revised.

  1. Serotonin is the most important chemical in this study, but authors did not mention the chemical source of serotonin in M&M.

Reply: It has been revised in Section 2.3.

  1. Table 1 should be next to 2.4 of M&M, and the accession number of each gene should be provided.

Reply: The accession number of each gene has been added as required, and Table 1 has been placed after 2.4 of materials and methods.

  1. Line 100, how did authors know the seedlings treated with 200 per was the best? According to Figure 1, I think seedlings treated with 100 and 200 per were similar.

Reply: According to figure 1, the seedlings treated with 100 and 200 µmol/L serotonin were similar, but the leaves of plants treated with 200 µmol/L serotonin were relatively dense. It has been revised in Section 3.1.

  1. Decimal places should be revised moderately in Table 2.

Reply: Because the unit of root length and stem length is cm, two decimal places were reserved. The weight was measured with a balance of one in ten thousand, and four decimal places were reserved.

  1. Line 117, should be H2O2.

Reply: It has been revised.

  1. SPAD value is just a value without any unit in Figure 4.

Reply: It has been revised in the figure.

  1. Line 147, proline content of the control group was significantly lower than that with 50 or 100 serotonin.

Reply: It is true that there is an error in expression. Actually, it is intended to express “There were no distinct differences between the proline content of the NaCl and that with 50 or 100 µmol/L serotonin”. It has been revised in this paper.

  1. Authors have to conduct statistical analysis for data of gene expression and mark significant differences in Figure 11-13.

Reply: It has been revised based on the suggestion.

  1. Figure 11, should be 200Ser

Reply: It has been revised based on the suggestion.

  1. Line 201, Arabidopsis should be in italic.

Reply: It has been revised based on the suggestion.

  1. The discussion of this manuscript is short and weak. Authors should cite more previous studies to support their results, hypothesis, or theory.

Reply: It has been revised based on the suggestion.

  1. Line 226, I didn't figure out which two ways.

Reply: It has been rewritten in the revised manuscript.

  1. Figure 2-4, 5-7, 8-10, and 11-13 would be combined as a figure with multiple panels, respectively.

Reply: It has been revised based on the suggestion.

Reviewer 3 Report

The paper entitled “Exogenous serotonin improves salt tolerance in rapeseed (Brassica napus L.) seedlings”, by Lui and coworkers, shows the positive effect of serotonin application on plants grown under salt stress. To demonstrate this, Lui and coworkers have analyzed different phenotypes such as dry and fresh root and shoot weight, chlorophyll accumulation, and accumulation of soluble sugar and protein. In all cases, serotonin application partially reverts to control condition growth the negative effects of NaCl. Moreover, this beneficial effect might be caused by the up-regulation of expression and activity of antioxidant enzymes. Although results are clearly exposed and indicate the role of serotonin as a salt stress modulator in B. napus, next I point out several comments that aim to improve the manuscript.  
Regarding experiment design, a control using only serotonin application (200µmol/L, since it’s the optimal concentration found) without salt stress would help understand its role in Brassica napus plants development, as it has been demonstrated for other species. In other words, does serotonin play a role in canola growth only under stress conditions?
Figures are clear, but manuscript would be more readable if plots belonging to a specific section were combined in the same figure. For instance, figures 2, 3, and 4, could be grouped in one. Same comment for the rest of plots along the manuscript. 
In the methods section, an explanation of statistical treatment of data must be included. I wonder why authors use Duncan test (line 134), and do not use ANOVA for statistical analysis. 
Along the manuscript, authors use “up-regulate” and “down-regulate” when they compare different measurement with control plants. The use of “increase” and “decrease” would be more correct. 
Line 107, percentage numbers indicated in the text should be revised, according to the data in table 2. In any case, authors compare plants under each concentration of serotonin under NaCl stress, with plants under salt stress. Under my point of view, it would be more correct to compare the different treatments with control plants. Moreover, authors compare percent values along the manuscript. It would be clearer if they name differences by “fold change”. This is applicable for the whole manuscript.
Overall, the results in different sections should be described in a coherent way all the time. For example, in section 3.4 (lines 141 to 149), authors do not mention at all any data for SS, but they give a whole explanation for data regarding SP. A similar trend is found in the results for MDA and H2O2 (lines 118-126). If there is a reason for that, please state it. 
Figure 2. The figure caption says “FW, fresh weight” (line 129), but it does not appear in the figure at all. This should be clarified.  Also, this figure caption indicates all the treatments performed for the experiment in this figure. It is assumable, as indicate in the manuscript, the rest of the figures keep these growth conditions. In figures 8 and beyond, however, it is not indicated anywhere whether serotonin treatment is combined with NaCl stress. Please, clarify this somewhere in the manuscript. 
In Figure 12, black bars represent 200 HT. Please correct or clarify this legend. 
In discussion (line 227-228), authors state that “serotonin does not rely on the receptor for free radical scavenging and antioxidant”. However, although serotonin seems to be an activator of antioxidant enzymes activity and osmolyte accumulation according to data provided in this work, data shown in this manuscript does not rule out the possibility of serotonin as a free radical scavenger. 
Finally, in the discussion section it would be appreciated to discuss the relevance of this findings. Do the authors find possible the use of such a chemical compound to improve plant growth under field conditions?

Author Response

  1. Regarding experiment design, a control using only serotonin application (200µmol/L, since it’s the optimal concentration found) without salt stress would help understand its role in Brassica napus plants development, as it has been demonstrated for other species. In other words, does serotonin play a role in canola growth only under stress conditions?

Reply: In our study, we only performed the research under stress conditions. Whether serotonin can promote the growth of rape under normal conditions needs further research.

  1. Figures are clear, but manuscript would be more readable if plots belonging to a specific section were combined in the same figure. For instance, figures 2, 3, and 4, could be grouped in one. Same comment for the rest of plots along the manuscript. 

Reply: It has been revised based on the suggestion.

  1. In the methods section, an explanation of statistical treatment of data must be included. I wonder why authors use Duncan test (line 134), and do not use ANOVA for statistical analysis. 

Reply: The data was analyzed by one-way variance method, but Duncan wass chosen in the assumption of homogeneity of variance. The detailed description of data statistical processing has been added in material method 2.5.

  1. Along the manuscript, authors use “up-regulate” and “down-regulate” when they compare different measurement with control plants. The use of “increase” and “decrease” would be more correct. 

Reply: It has been revised based on the suggestion.

  1. Line 107, percentage numbers indicated in the text should be revised, according to the data in table 2. In any case, authors compare plants under each concentration of serotonin under NaCl stress, with plants under salt stress. Under my point of view, it would be more correct to compare the different treatments with control plants. Moreover, authors compare percent values along the manuscript. It would be clearer if they name differences by “fold change”. This is applicable for the whole manuscript.

Reply:Thanks for the reviewer’s opinions. The percentage numbers were carefully revised. As shown in Table 2, all the characters were significantly reduced by salinity whenever with or without serotonin. Therefore, percentage values were used to display the differences.

6.Overall, the results in different sections should be described in a coherent way all the time. For example, in section 3.4 (lines 141 to 149), authors do not mention at all any data for SS, but they give a whole explanation for data regarding SP. A similar trend is found in the results for MDA and H2O2 (lines 118-126). If there is a reason for that, please state it. 

Reply: In the manuscript, we only described the data trend of SP. The other substance had a similar change trend, the content of which reached the highest value when treated with 200 µmol/L serotonin, and showed a downward trend with 300 µmol/L serotonin. Therefore, they were not described one by one.

7.Figure 2. The figure caption says “FW, fresh weight” (line 129), but it does not appear in the figure at all. This should be clarified.

Reply: At first, FW was used in the legend, so it was explained in detail in the annotation. Later, the graph was modified many times, and we forgot to delete it. This has been modified in the revised manuscript.

 8.Also, this figure caption indicates all the treatments performed for the experiment in this figure. It is assumable, as indicate in the manuscript, the rest of the figures keep these growth conditions. In figures 8 and beyond, however, it is not indicated anywhere whether serotonin treatment is combined with NaCl stress. Please, clarify this somewhere in the manuscript. 

Reply: A detailed description has been added to each diagram, and the combination of 5-HT treatment and NaCl stress is explained in details.

  1. In Figure 12, black bars represent 200 HT. Please correct or clarify this legend. 

Reply: It has been revised based on the suggestion.

10.In discussion (line 227-228), authors state that “serotonin does not rely on the receptor for free radical scavenging and antioxidant”. However, although serotonin seems to be an activator of antioxidant enzymes activity and osmolyte accumulation according to data provided in this work, data shown in this manuscript does not rule out the possibility of serotonin as a free radical scavenger. 

Reply: It has been revised based on the suggestion.

  1. Finally, in the discussion section it would be appreciated to discuss the relevance of this findings. Do the authors find possible the use of such a chemical compound to improve plant growth under field conditions?

Reply: At present, only indoor studies have been done on this compound. In the future, it is possible to study whether it can improve plant growth in fields.

Round 2

Reviewer 2 Report

  1. Line 47-51, needs some papers to support thesis sentences “Under salt stress, Osmolytes, such as sugars and protein, are accumulated and external osmotic pressure are decreased. In order to maintain the balance of osmotic pressure in cells, osmotic adjustment substances such as solid sugar (SS), solid protein (SP) and proline (pro) are accumulated in plants to continuously absorb water from the environment and prevent water leakage”. 
  2. Furthermore, I think the following should be deleted “ to continuously absorb water from the environment and”.
  3. Line 50, SS and SP should be soluble sugar and soluble protein, respectively.
  4. Line 74, Helianthus annuus should be in italic.
  5. Where is the hypothesis of this study?
  6. In figure 2, chlorophyll content was determined by SPAD. SPAD value is just a value without a unit. Please delete the unit of chlorophyll content in figure 2C.
  7. In section 3.5, “up-regulated” and “down-regulated” are moderate to describe the gene expression of POD7, CAT3, and Cu-SOD.
  8. Line 271-274,  need some papers to support thesis sentences “With the increase of concentration, the alleviating effect of serotonin on salt stress gradually increased, but when the concentration reached 300 μmol/L, the alleviating effect began to weaken obviously, which indicated that high concentration of serotonin not only could not alleviate salt stress, but also could produce ion toxicity to plants.”
  9. Line 276-277, needs some papers to support thesis sentences “Salinity affected the photosynthetic capacity of photosynthetic organs and decreased photosynthetic characters in plants.
  10. Line 292-293, I don’t figure out these following words, “and even makes it a wrong physiological metabolic molecule.”
  11. Line 296-296, authors should cite some previous studies to support the following, “It was also indicated that the membrane lipid peroxidation of plants was enhanced under salt stress, resulting in the increase of the MDA content. The addition of serotonin alleviated the membrane lipid peroxidation under salt stress.”
  12. Line 313-316, please cite some previous studies to support this theory “This study indicated that there are two ways for serotonin in protecting the plants from injuries induced by oxidative stress under salinity. On the one hand, serotonin might act as a free radical scavenger indepdent of the antioxidant system. On the other hand, it can enhance the gene expression and the activity of antioxidant enzymes.” 

Reviewer 3 Report

Authors have included some of the changes that me and other reviewers have proposed, and have assesed the questions I have made in my previous review. For this reasons, I thank Liu and coworkers. I will point out several formal aspects that might be considered for final publications of the manuscript. 
In abstract line 22, variations in gene expression is normally referred as up-regulated or down-regulated. Some other examples of gene expression levels are found in lines 222, 225, 226, and others. 
Line 45. “in enzymatic defense syste”. Revise. 
Line 74. “Helianthus annuus” must be italic. 
Figure 1. Why authors use CK to indicate control conditions, whereas in the rest of the manuscript is used C? I would recommend using C for the whole manuscript. 
Line 164. Increased H2O2 value indicated is 26.9%. According to graph in Figure 2, accumulation of H2O2 seems much higher. Can it be circa 126% instead? The exact value is not possible to be calculated, since raw data are not shown. 
A general trend in data shown in the manuscript, it seems that authors compare salinity conditions with control conditions, and serotonine + salinity conditions are compared to salinity conditions. If this is the case, it should be clarified each time. What is the percentage value compared to? As examples, in line 166 percentage values are shown, and I assume it is the decrease of H2O2 accumulation upon serotonine application under salt conditions. If this is the case, it should be clarified. Some other example is the percentage values shown in line193. In some other cases (enzymatic activities for instance, lines 215-216), it is not clear whether percentage values of activity upon serotonine treatment are compared to C or N condition. 
Line 168-169. For SPAD values, I would better use a biological parameter, such as chlorophyll accumulation. 
Line 191. “There were not distinct differences” is redundant. I recommend to eliminate the word “distinct” of this sentence. 
